# The Role of Hsp90-R2TP in Macromolecular Complex Assembly and Stabilization

**DOI:** 10.3390/biom12081045

**Published:** 2022-07-28

**Authors:** Jeffrey Lynham, Walid A. Houry

**Affiliations:** 1Department of Biochemistry, University of Toronto, Toronto, ON M5G 1M1, Canada; jeffrey.lynham@mail.utoronto.ca; 2Department of Chemistry, University of Toronto, Toronto, ON M5S 3H6, Canada

**Keywords:** molecular chaperones, Hsp90, R2TP, PAQosome, TTT, snoRNP, snRNP, RNA polymerase, PIKK, TSC, dynein arm

## Abstract

Hsp90 is a ubiquitous molecular chaperone involved in many cell signaling pathways, and its interactions with specific chaperones and cochaperones determines which client proteins to fold. Hsp90 has been shown to be involved in the promotion and maintenance of proper protein complex assembly either alone or in association with other chaperones such as the R2TP chaperone complex. Hsp90-R2TP acts through several mechanisms, such as by controlling the transcription of protein complex subunits, stabilizing protein subcomplexes before their incorporation into the entire complex, and by recruiting adaptors that facilitate complex assembly. Despite its many roles in protein complex assembly, detailed mechanisms of how Hsp90-R2TP assembles protein complexes have yet to be determined, with most findings restricted to proteomic analyses and in vitro interactions. This review will discuss our current understanding of the function of Hsp90-R2TP in the assembly, stabilization, and activity of the following seven classes of protein complexes: L7Ae snoRNPs, spliceosome snRNPs, RNA polymerases, PIKKs, MRN, TSC, and axonemal dynein arms.

## 1. Overview of Hsp90 Structure and Its Function with R2TP

The Hsp90 molecular chaperone is a central regulator of protein homeostasis in eukaryotes under normal and stressed conditions. Hsp90 is involved in the final stages of client protein folding and maturation. In mammals, there are two cytoplasmic Hsp90 isoforms, Hsp90α and Hsp90β, while in yeast, Hsp82 and Hsc82 are the inducible and constitutively expressed Hsp90 isoforms, respectively [1]. Hsp90 isoforms (referred to here as Hsp90) exist as dynamic homodimers, with each protomer comprised of three domains: an N-terminal domain, the site of ATP binding and hydrolysis [2]; a middle domain, which interacts with Hsp90 substrates; and a C-terminal domain, which forms the Hsp90 dimerization interface (Figure 1A) [3]. The C-terminal domain also contains a MEEVD motif, which is important for interactions with Hsp90 cochaperones that contain TPR domains (see Table 1 for nomenclature). Hsp90 substrates are called clients, and the current set of Hsp90 clients includes steroid hormone receptors, kinases, transcription factors, E3 ubiquitin ligases, and many others that share no common features in terms of sequence, structure, or function [4]. Hsp90-mediated client folding and stabilization is a regulated process that requires the association and release of chaperones and cochaperones. Hsp90 client loading is largely dependent on Hsp70, which binds to nascent or partially folded polypeptides with exposed hydrophobic residues [5,6], and Hop, which functions as an adaptor between Hsp70 and Hsp90 [7].

In addition to stabilizing tertiary structure, Hsp90 and its cochaperones stabilize the quaternary structure of various macromolecular complexes. In 2005, our group identified Tah1 and Pih1 as Hsp90 interactors in yeast [8]. Tah1 and Pih1 form a heterodimer and interact with AAA+ proteins Rvb1 and Rvb2 to form the R2TP chaperone complex that is conserved in higher eukaryotes including humans. Most notably, the R2TP complex is involved in the assembly of L7Ae ribonucleoproteins [9,10,11], RNA polymerases [12], and PIKK complexes [13]. In humans, R2TP associates with RNA polymerase subunit RPB5, WD40 repeat protein WDR92, and the Unconventional Prefoldin Complex (UPC), comprising of URI1, UXT, PDRG1, PFDN2, PFDN6, and ASDURF [14,15,16]. Altogether, these 12 proteins constitute the PAQosome, Particle for Arrangement of Quaternary Structure (Figure 1A) [17]. The PAQosome is the largest and most intricate chaperone interacting with Hsp90. The R2TP complex is involved in all PAQosome-mediated pathways as the catalytic component, whereas the function of the other subunits is mostly unknown. WDR92 has a specialized role in dynein arm assembly [18], RPB5 likely bridges the interactions between the PAQosome and RNA polymerases, and the UPC may regulate R2TP in response to cell growth and proliferation [19]. Moreover, URI1 mediates nuclear and cytoplasmic shuttling of RNAP subunits, and it has been suggested to do so as part of the PAQosome [20,21]. Thus, PAQosome assembly may occur in the cytoplasm with URI1 facilitating its transport into the nucleus and vice-versa (Figure 2).

Within the PAQosome, RPAP3 and PIH1D1 are proposed to function as scaffolds for Hsp90 and its diverse client proteins. RPAP3 contains an RPAP3_N domain that mediates interactions with substrates enriched with helical-type domains [22]; two TPR domains, whereby TPR2 has high affinity for Hsp90 [23]; an intrinsically disordered region that makes contacts with RUVBL1 [22]; and an RPAP3_C domain that binds to the ATPase side of RUVBL2 [24]. PIH1D1 contains an N-terminal PIH1 domain that binds DpSDD/E motifs on clients [25,26] and a C-terminal CHORD and Sgt1 (CS) domain that binds RPAP3 [24,27]. Although it has been proposed that PIH1D1 binds to and regulates RUVBL2 ATPase activity as a nucleotide exchange factor, our group has shown that, within the R2TP complex, PIH1D1 binds exclusively to RPAP3 and that PIH1D1 has little effect on RUVBL1/2 ATPase activity and nucleotide binding affinity [22]. Interestingly, although our model suggests that PIH1D1 only interacts with RPAP3 within the R2TP complex, we have identified R2T and R2P complexes in vitro and *in cellulo* [22]. The significance of these findings in regard to Hsp90 function has yet to be determined.

In yeast, Tah1 is much smaller than RPAP3 and contains two TPR repeats followed by a C-helix and an unstructured region [28,29]. The TPR domain binds the Hsp90 C-terminal MEEVD motif, while the unstructured region binds Pih1. Yeast Pih1 is slightly larger than PIH1D1 and contains an N-terminal PIH1 domain, which also recruits clients with DpSDD/E motifs, and a C-terminal CS domain that binds Tah1 [26,28,30]. The Tah1-Pih1 dimer binds to the Rvb1/2 hexamer DII domains to form the R2TP complex [31,32]. Yeast R2TP forms an open basket that accommodates client proteins and Hsp90 (Figure 1B).

Although it has been established that human Hsp90 interacts with R2TP through RPAP3 [23,24], the details of Hsp90-mediated protein complex assembly are limited, with most of our knowledge restricted to proteomics and in vitro interaction analyses. This review will discuss our current understanding of Hsp90-R2TP in higher metazoans and its roles in protein complex assembly, stabilization, function, or localization for seven classes of protein complexes: L7Ae snoRNPs, spliceosome snRNPs, RNA polymerases, PIKKs, MRN, TSC, and axonemal dynein arms (Figure 2). Of note, human Hsp90 in these studies may refer to either isoform, Hsp90α or Hsp90β, since they have nearly identical structural and functional similarities that cannot be easily distinguished from one another.

## 2. snoRNP Biogenesis

Eukaryotic ribosomal RNA (rRNA) processing occurs in the nucleolus, which contains numerous small nucleolar RNAs (snoRNA). Most snoRNAs function as sequence-specific guides during rRNA modification [33], while others are involved in folding and cleavage events [34,35]. There are two major families of snoRNAs: box C/D and box H/ACA. Box C/D snoRNAs direct ribose 2′-O-methylation within rRNA and certain spliceosome small nuclear RNA (snRNA) [36,37], and box H/ACA snoRNAs direct the isomerization of uridine to pseudouridine [38]. They are classified based on conserved sequence motifs and their association with common core proteins. Mature snoRNP complexes are comprised of snoRNA and four common core proteins, namely, fibrillarin, NOP56, NOP58, and 15.5K for box C/D snoRNPs, and NHP2, NOP10, GAR1, and NAP57 for box H/ACA snoRNPs (Figure 3). During snoRNP biogenesis, Hsp90 stabilizes NOP58, 15.5K, and NHP2 [9].

### 2.1. Box C/D snoRNP Assembly

#### 2.1.1. Role of Hsp90 in Box C/D snoRNP Assembly

Regardless of their snoRNA component, box C/D snoRNPs have a highly conserved asymmetric arrangement of core proteins [39,40,41,42]. The 15.5K protein is part of the L7Ae family of ribosomal proteins and was first identified as a component of the U4/U6.U5 tri-snRNP that binds directly to the 5′ stem loop of U4 snRNA [43], which has a similar primary and secondary structure to box C/D and C′/D′ motifs [44]. 15.5K binding to the box C/D motif is essential for the recruitment of assembly factors RUVBL1 and RUVBL2 and other core proteins including NOP56, NOP58, and fibrillarin [45]. NOP56 and NOP58 are two paralogous proteins that contain NOP and coiled-coil (CC) domains [46]. The NOP domain exhibits RNA and protein binding, which allows NOP56 and NOP58 binding to 15.5K-box C/D snoRNA complexes [47], while the CC domain enables NOP56-NOP58 heterodimerization across the C/D and C′/D′ motifs [39,42]. The NOP56 and NOP58 N-terminal domains together recruit one copy of fibrillarin to the snoRNP complex [39].

Box C/D snoRNP formation requires Hsp90. In HeLa cell extracts, Hsp90, RPAP3, and PIH1D1 coprecipitated with precursor and mature forms of ectopically expressed rat U3 snoRNA [9]. HEK293 cells expressing rat U3 snoRNA and treated with geldanamycin, an Hsp90 inhibitor that blocks the ATP binding site [2,48], had less U3 snoRNA accumulation [9]. These findings were the first to indicate a role for Hsp90 in box C/D snoRNP biogenesis.

Subsequent experiments have suggested that the role of Hsp90 in box C/D snoRNP biogenesis is to stabilize core protein NOP58. When HEK293 cell lines expressing GFP-tagged proteins were treated with geldanamycin, NOP58 and 15.5K failed to accumulate, while a mild effect was seen for NOP56 [9]. NOP58 mutants, NOP58-K310A-A313R, and NOP58-A283P, which cannot assemble into mature snoRNPs, showed stronger interactions with Hsp90 than NOP58-WT [49]. NOP58-A283P also associated with the Hsp70-Hsp90 adaptor Hop [49]. Therefore, NOP58 is likely stabilized through the Hsp70-Hop-Hsp90 pathway during its maturation and assembly into snoRNPs. Interestingly, Hsp90 also stabilized the L7Ae protein SBP2, suggesting that Hsp90 is also involved in SECIS mRNP biogenesis (Figure 3) [9]. SECIS mRNPs associate with selenoprotein mRNAs for translational recoding of a UGA codon that enables the insertion of selenocysteine [50].

#### 2.1.2. Role of R2TP in Box C/D snoRNP Assembly

NOP58 can be stabilized by other chaperones, namely the RUVBL1/2 complex and NOPCHAP1 [49]. RUVBL1 and RUVBL2 were identified as box C/D snoRNP biogenesis factors from an early study identifying mouse U14 snoRNA interactors [51]. Subsequent studies have shown that RUVBL1/2 interact with precursor and mature forms of rat U3 and human U8 snoRNA [9,52,53]. NOPCHAP1 was identified as a snoRNP assembly factor through Stable Isotope Labeling with Amino Acids in Cell Culture (SILAC) experiments, which showed that NOPCHAP1 and RUVBL1/2 associated with nascent NOP58 [54]. NOPCHAP1 binds to NOP58 through the CC-NOP fragment, while it binds to RUVBL1 through the DII domain [49]. The interaction between NOP58 and RUVBL1 is weak, but in the presence of NOPCHAP1, it is enhanced 20-fold [49]. RUVBL1 binds to NOPCHAP1 in the absence of ATP since ATPγS, a non-hydrolyzable ATP analogue, abolished NOPCHAP1 binding [49]. The interaction between NOPCHAP1 and RUVBL1/2 is likely transient and may serve only to direct NOP58 to RUVBL1/2. Interestingly, WT HEK293T cells treated with geldanamycin and NOPCHAP1 KO cells displayed similar levels of reduced NOP58, indicating that Hsp90 and NOPCHAP1 may act on the same pathway [49]. A caveat to consider is that geldanamycin may have additional binding targets that affect the viability of Hsp90 clients.

RUVBL1/2 may also act as a NOP58 chaperone as part of the R2TP complex. HeLa cell extracts separated on linear glycerol gradients showed the assembly factor NUFIP1 and core proteins NOP58 and fibrillarin to be in the same fractions as RUVBL1, RUVBL2, and RPAP3 [9]. Also, pulldown assays in rabbit reticulocyte lysates showed that PIH1D1 directly interacts with NOP56 and NOP58 [9], and that RUVBL1 and RUVBL2 interact with all four box C/D snoRNP proteins [55]. Moreover, the R2TP complex is involved in Cajal body and nucleolar localization of pre-snoRNPs and mature snoRNPs, respectively. In HeLa cells transfected with siRNA, depletion of RUVBL1 and RUVBL2 caused reductions of Cajal body and nucleolar U3 snoRNA [55].

Hsp90 inhibition in HEK293 cells resulted in the disappearance of both NOP58 and 15.5K [9], but the link between Hsp90 ATPase activity and 15.5K stabilization remains unclear. Hsp90 may indirectly stabilize 15.5K by stabilizing NOP58 first. Rather than interacting with Hsp90, 15.5K can bind RUVBL1, RUVBL2, and RUVBL1/2 in the presence of ATP [55,56]. Additionally, RUVBL1/2 was shown to bridge the interaction between 15.5K and core proteins NOP56 and NOP58 [55], which may be important for 15.5K stability. Taken together, Hsp70, Hop, Hsp90, R2TP, and NOPCHAP1 stabilize NOP58, and RUVBL1/2 subsequently recruits 15.5K to NOP56-NOP58, thereby stabilizing 15.5K.

#### 2.1.3. R2TP-Associated Box C/D snoRNP Assembly Factors

The R2TP complex is a highly interactive chaperone complex that works together with other box C/D snoRNP assembly factors, namely, NUFIP1, ZNHIT3, and ZNHIT6 [54]. NUFIP1 acts mainly as a tethering protein, joining 15.5K with NOP56, NOP58, and fibrillarin [9,56]. However, NUFIP1 may also regulate R2TP function during snoRNP assembly since it interacts directly with RUVBL1, RUVBL2, and PIH1D1 [9,56]. For example, the coprecipitation of 15.5K with either NOP56 or NOP58 was enhanced with RUVBL1/2, but was repressed with both RUVBL1/2 and NUFIP1 [55].

NUFIP1 forms a heterodimer with ZNHIT3, an assembly factor belonging to the zf-HIT family, which are often observed in complexes containing RUVBL1 and RUVBL2 [54,57,58,59,60]. ZNHIT3 is required for NUFIP1 stability since siRNA-mediated depletion of ZNHIT3 in HeLa cells resulted in similar decreases in NUFIP1 levels [60]. ZNHIT3 was unable to bind precursor or mature forms of rat U3 snoRNA, but it was able to bind U3 snoRNA mutants that had decreased affinity for NOP56, NOP58, and fibrillarin [54]. 

Finally, in the presence of ATP, ZNHIT6 interacts with the RUVBL1/2 complex, but not with individual RUVBL1 and RUVBL2 proteins [55]. In addition, ZNHIT6 binds 15.5K, but not NOP56, NOP58, or fibrillarin [56].

### 2.2. Box H/ACA snoRNP Assembly

The assembly of box H/ACA snoRNPs has been well-established. The pseudouridine synthase NAP57 (also named dyskerin) and core proteins NOP10 and NHP2 form a trimer that binds directly to H/ACA RNA in the absence of GAR1 [61]. Early yeast genetic depletion studies have demonstrated that the core trimer is required for H/ACA RNA stability and that all four core proteins are essential for cell viability [62,63,64,65,66]. The assembly of mammalian H/ACA snoRNPs requires two assembly factors, NAF1 and SHQ1, which are needed for H/ACA RNA accumulation without being part of the mature particles [67,68]. NAF1 is structurally similar to GAR1 [69]. During snoRNP biogenesis, NAF1 and the core trimer associate with H/ACA RNA at the site of transcription [70]. Upon snoRNP maturation, snoRNP particles localize to Cajal bodies or nucleoli where GAR1 replaces NAF1 [68,70]. SHQ1 functions as a NAP57 chaperone by acting as an RNA placeholder, thereby protecting NAP57 from nonspecific RNA binding before its association with H/ACA RNA and other core RNP proteins [71].

#### 2.2.1. Role of Hsp90 in Box H/ACA snoRNP Assembly

Hsp90 is involved in H/ACA snoRNP biogenesis since Hsp90 inhibition led to defects in H/ACA RNA production and core protein stability [9]. HEK293 cells treated with geldanamycin showed decreased levels of telomerase H/ACA RNA [9], which is consistent with TERT, the reverse transcriptase in the telomerase complex, being an Hsp90 client [72]. In addition, geldanamycin-treated cells showed a complete loss of core protein NHP2, indicating NHP2 as a potential Hsp90 client [9]. NAP57 levels were unaffected by geldanamycin [9], presumably because it was stabilized by SHQ1 [73].

Telomerase is a box H/ACA snoRNP complex that synthesizes the G-rich DNA at the 3´-ends of linear chromosomes [74]. In addition to the four box H/ACA core proteins, human telomerase contains reverse transcriptase TERT and telomerase RNA component TERC (Figure 3). Hsp90 and its cochaperone p23 bind TERT, and blocking this interaction inhibits the proper assembly of active telomerase in vitro [72]. TERT and TERC could bind to each other without Hsp90-p23, although this complex was inactive [75]. In addition, Hsp90 inhibitors geldanamycin and novobiocin inhibited telomerase even after telomerase was assembled [75]. Unlike most of their clients, Hsp90 and p23 remain associated with active telomerase [76]. In mammalian cells, Hsp90 regulates TERT expression. In SCC4 cells, a telomerase-positive oral cancer cell line, coprecipitation experiments showed an in vivo interaction between Hsp90 and the TERT promoter. Geldanamycin exposure decreased telomerase activity, TERT promoter activity, and TERT mRNA expression [77]. Additionally, in cerebral endothelial cells, siRNA-mediated Hsp90 depletion inhibited telomerase activity and decreased telomerase protein expression [78].

#### 2.2.2. Role of R2TP in Box H/ACA snoRNP Assembly

Hsp90 chaperone function on telomerase may depend on R2TP, since RUVBL1 and RUVBL2 were reported to interact with TERT and NAP57 [79]. During telomerase assembly, RUVBL1 and RUVBL2 may stabilize NAP57 since depletion of RUVBL1 and RUVBL2 led to a significant reduction of NAP57 steady-state levels [79]. Moreover, RUVBL1 and RUVBL2 associated with TERC in HeLa cell extracts, and RUVBL1 ATPase activity was essential for TERC maintenance [79]. In addition, RUVBL1 and RUVBL2 are involved in the production of other H/ACA RNAs. RUVBL1- or RUVBL2-siRNA knockdown in HeLa cells caused a reduction in the H/ACA RNAs E3 and U17/E1 [10].

Regarding H/ACA snoRNP complex assembly, knockdown of RUVBL1 and RUVBL2 in HeLa cells resulted in a loss of NHP2 and NAP57 [10]. NHP2 makes a direct interaction with NUFIP1 [9], suggesting that NUFIP1 could bridge the interaction between NHP2 and RUVBL1/2 to mediate NHP2 assembly or stability. Similar to its role in box C/D snoRNPs, NUFIP1 may also be involved in bridging interactions between H/ACA snoRNA and core proteins. NUFIP1 coprecipitated with U19 H/ACA snoRNA, and its depletion reduced the levels of U19 and telomerase RNA [9].

During snoRNP assembly, NAP57 is stabilized by the RUVBL1/2 complex and SHQ1 [73,79], and NAP57 requires the R2TP complex to dissociate from SHQ1 [10]. SHQ1 exerts a clamp-like grip on NAP57 through binding to NAP57 *in trans*: the n-terminal CS domain of SHQ1 binds to the surface that is opposite from the RNA binding surface where the c-terminal SHQ1-specific domain binds [10]. NAP57 recruits the R2TP complex through its unstructured C-terminus [10]. RUVBL1, RUVBL2, and PIH1D1 bind to the same domain on NAP57 as SHQ1. RUVBL1 and RUVBL2 also bind to the CS domain of SHQ1 and remove it from NAP57 through an ATP-independent mechanism [10]. ATP binding and hydrolysis may only be required for the release of RUVBL1/2 from NAP57 after SHQ1 has been removed.

## 3. Spliceosome snRNP Assembly

The spliceosome is a molecular machine that catalyzes splicing, an essential post-transcriptional modification that removes introns from pre-mRNA. Spliceosomes are comprised of the Prp19 complex, U1 snRNP, U2 snRNP, and the U4/U6.U5 tri-RNP, with each snRNP having their own snRNA component and associated proteins. The spliceosome associates with more than 300 different proteins [80]. Proteomic analyses of purified spliceosomes have shown that the complex is highly conserved, with more than 85% of yeast proteins having a direct human orthologue [81]. Proteomic analyses in yeast and human cells have revealed a role for Hsp90 and R2TP in U4 and U5 snRNP assembly [54,58,59,82]. 

### 3.1. U4 snRNP Assembly

#### 3.1.1. Similarities between U4 snRNP and Box C/D snoRNP Complexes

The U4 snRNP is comprised of U4 snRNA, L7Ae protein 15.5K, and splicing component PRPF31 (Figure 3). In addition, 15.5K binds to the 5′ stem-loop of U4 snRNA in a manner similar to the box C/D motif [43,45], enabling PRPF31 recruitment [83]. Without 15.5K, PRPF31 weakly associates with U4 RNA [82]. The association between PRPF31 and 15.5K is essential because an A216P mutation in PRPF31, which abolished the PRPF31-15.5K interaction, resulted in PRPF31 cytoplasmic accumulation, indicating the prevention of PRPF31 incorporation into mature spliceosomes within the nucleus [82].

U4 snRNP shares a few similarities with box C/D snoRNP. In addition to both containing the RNA binding component 15.5K, the U4 snRNP splicing component PRPF31 is homologous to box C/D core proteins NOP56 and NOP58. These three proteins each have a NOP domain, which binds to preformed 15.5K-RNA complexes, and a CC domain, which mediate protein–protein interactions within RNP complexes [47]. Furthermore, U4 snRNA and box C/D snoRNA both associate with the assembly factor NUFIP1, but unlike box C/D snoRNA, U4 snRNA is not dependent on NUFIP1 for its assembly and maturation [9].

#### 3.1.2. Role of Hsp90 and R2TP in U4 snRNP Assembly

Similar to box C/D snoRNP assembly, U4 snRNP assembly and stabilization is mediated by chaperones R2TP, Hsp90, and Hsp70. Co-IP experiments using antibodies against Hsp90, RUVBL1, RUVBL2, RPAP3, and PIH1D1 showed that each associated with U4 snRNA [9]. HEK293 cells treated with geldanamycin showed an almost complete loss of U4 snRNA, a moderate decrease of 15.5K, and a mild effect on PRPF31 [9]. In PRPF31, an A216P mutation prevents its nuclear localization [84], and a K243A/A246R double mutation prevents its interaction with 15.5K [82]. SILAC-IP experiments using HeLa cells expressing PRPF31-A216P or PRPF31-K243A/A246R showed that both PRPF31 mutants were enriched with Hop and Hsp70 [49]. Hsp90 was also present at low levels [49]. These findings show that PRPF31 binds to Hsp70 in the cytoplasm and suggest that the Hsp70-Hop-Hsp90 pathway mediates the PRPF31-15.5K interaction.

The assembly factors NUFIP1 and ZNHIT3 together with the R2TP complex are also involved in U4 snRNP biogenesis. In mammalian cells, NUFIP1 can mediate the interaction between 15.5K and PRPF31, and it may do so with help from the R2TP complex since NUFIP1 also binds to RUVBL1, RUVBL2, and PIH1D1 [9,56]. Moreover, coprecipitation experiments in HEK293T cells showed that NUFIP1, ZNHIT3, and RUVBL1 each associate with U4 snRNA and PRPF31, suggesting that they can mediate the interaction between U4 snRNA and PRPF31 [82]. Indeed, NUFIP1 knockout cells had a two-fold reduction in binding between PRPF31 and U4 snRNA [82].

### 3.2. U5 snRNP Assembly

#### 3.2.1. Role of Hsp90 and R2TP in U5 snRNP Assembly

U5 snRNP is recruited to the spliceosome as part of the U4/U6.U5 tri-snRNP and is comprised of U5 snRNA, GTPase EFTUD2, helicase SNRNP200, and mRNA processing factor PRPF8 (Figure 3). The Hsp90/R2TP complex is mostly involved in the stabilization and assembly of PRPF8 into mature U5 snRNP particles. SILAC experiments showed that Hsp90 associates with PRPF8 and EFTUD2, as well as with assembly factors AAR2 and ECD [59]. In HeLa cells, Hsp90 ATPase activity was shown to stabilize PRPF8 and SNRNP200, but not EFTUD2 [59]. Hsp90 was also shown to mediate the interaction between PRPF8 and cytoplasmic RPAP3 [59].

PRPF8 is stabilized by the R2TP complex and U5 snRNP assembly factors. In vitro pulldown experiments have shown FLAG-tagged PRPF8 to simultaneously co-elute with purified RUVBL1-RUVBL2, RPAP3-PIH1D1, AAR2, ECD, and ZNHIT2 [85]. PRPF8 can make direct interactions with RUVBL1-RUVBL2 and RPAP3-PIH1D1. The PRPF8-RUVBL1/2 interaction is stronger than the PRPF8-RPAP3-PIH1D1 interaction [85]. PRPF8 mutants that cannot be integrated into mature U5 snRNPs associate more strongly with R2TP than WT PRPF8 [59]. PRPF8 binding to R2TP and AAR2 chaperones was increased in the absence of PIH1D1, suggesting that the formation of the R2TP complex through PIH1D1 binding is important for the release of PRPF8 from R2TP and AAR2 [59].

EFTUD2 may also be stabilized by the R2TP complex and AAR2. EFTUD2 has a DSDED motif, suggesting that it binds to PIH1D1; however, the PIH1D1 N-terminal domain was not sufficient to bind EFTUD2 [59]. Although, mutations in the EFTUD2 DSDED motif did affect EFTUD2 binding with AAR2, RUVBL1/2, and ZNHIT2 [59]. In addition, when PIH1D1, RUVBL2, and ZNHIT2 were depleted, there was less EFTUD2 [59]. EFTUD2 may be recruited to the R2TP complex through RPAP3 since EFTUD2 was shown to interact with ectopically expressed FLAG-RPAP3 in HEK293 lysates [58].

#### 3.2.2. R2TP-Associated U5 snRNP Assembly Factors

ZNHIT2 and the R2TP complex mediate U5 snRNP subunit interactions during U5 snRNP assembly. SILAC experiments showed that ZNHIT2 interacts with R2TP, EFTUD2, PRPF8, SNRNP200, and yeast two-hybrid experiments confirmed direct interactions between ZNHIT2-EFTUD2 and ZNHIT2-RUVBL1 [59]. When ZNHIT2 is knocked out, the interactions between RPAP3-EFTUD2 and RPAP3-PRPF8 were absent [58]. ZNHIT2 is also needed to bridge the binding of RUVBL1/2 with EFTUD2 and PRPF8 [58]. The RUVBL1/2-ZNHIT2 cryo-EM structure shows that the RUVBL1/2 DII domains interact with the ZNHIT2 C-terminal end [85], which is in contrast to another study that showed that the ZNHIT2 HIT domain was essential for binding [58]. Rather than mediating the RUVBL1/2-ZNHIT2 interaction, the HIT domain may regulate the conformation and nucleotide state of RUVBL1/2 [85]. Through binding to the DII domains, ZNHIT2 disrupts the RUVBL1/2 dodecamer [85]. When ZNHIT2 was bound to the hexamer, RUVBL1 still had ADP bound while RUVBL2 was in the apo state [85]. Interestingly, the intrinsically low ATPase activity of RUVBL1/2 hexamers with one Walker B mutant, in either RUVBL1 or RUVBL2, was significantly increased with ZNHIT2 present, suggesting that ZNHIT2 affects the activity of both RUVBL1 and RUVBL2 subunits [85].

ECD is another adaptor protein involved in U5 snRNP biogenesis. In vitro pulldowns showed that ECD co-eluted with GST-ZNHIT2 and RUVBL1/2, and its association with this complex was enhanced when RPAP3 and PIH1D1 were added [85]. ECD can bind RUVBL1 and PIH1D1, either through its DpSDD motif or another uncharacterized binding site [25,86].

## 4. Hsp90- and R2TP-Mediated RNA Polymerase Assembly and Localization

The eukaryotic RNA polymerases, RNAP I, RNAP II, and RNAP III, are multiprotein complexes that synthesize ribosomal, messenger, and transfer RNA, respectively. The three RNA polymerases are structurally related. Within each complex, the two largest subunits form the catalytic core, while the smaller subunits are located on the periphery. They are also related through having five common subunits: RPB5, RPB6, RPB8, RPB10, and RPB12. Large-scale proteomic screens identified Hsp90, R2TP, and prefoldins as RNAP II interactors (Figure 1) [14,16,87,88]. RNAP II is assembled in the cytoplasm by Hsp90 and R2TP and then imported into the nucleus through URI1 [12,20]. To further analyze the interactions of RNAP II subunits during assembly, Boulon and colleagues performed triple-SILAC purifications on U2OS cells treated with α-amanitin, a small molecule that binds RPB1 and induces its degradation [12,89]. Their findings revealed the presence of two subcomplexes: RPB1-RPB8 and RPB2-RPB3-RPB10-RPB11-RPB12. In addition, each subcomplex associated with a specific set of assembly factors, such as RPAP2, GPN2, GPN3, and GrinL1A. RPB1-RPB8 also associated with R2TP/Prefoldin components RPAP3, PFDN2, and UXT [12].

RPB1 is the largest subunit in RNAP II and interacts with many RNAP II subunits and assembly factors. Coprecipitation and yeast two-hybrid experiments showed that Hsp90 interacts with RPB1 and with the TPR2 domain on RPAP3 [12]. RPB1 interacts with RPAP3 outside of the TPR2 domain [12], implying that RPAP3 stabilizes RPB1 by tethering the interaction between Hsp90 and RPB1. Indeed, long-term RPAP3 depletion in U2OS cells resulted in RPB1 loss [12]. Also, RPAP3 depletion resulted in RPB1 cytoplasmic accumulation in mouse intestinal epithelium cells and crypt base columnar stem cells [90]. Another study showed that RNAP II assembly in melanoma cells was dependent on RPB1 interacting with URI1 [91], but the role of the prefoldin-like module during RNAP II assembly is uncharacterized.

The depletion of RNAP subunits leads to the accumulation of unstable cytoplasmic RPB1. Boulon and colleagues showed that siRNA-mediated depletion of any RNAP II subunit in U2OS cells resulted in RPB1 cytoplasmic accumulation [12]. When cells were treated with geldanamycin, there was a significant decrease of RPB1 in RPB2-, RPB3-, RPB5-, RPB8-, RPB10-, RPB11-, and RPB12-depleted cells, but no significant changes of RPB1 in RPB4-, RPB6-, RPB7-, and RPB9-depleted cells [12]. Hsp90 is essential for stabilizing RPB1, however, Hsp90 binding to RPB1 occurred independent of its ATPase activity [12]. To stabilize RPB1, Hsp90 ATPase activity may mediate interactions between RPB1 and RNAP II subunits RPB5, RPB8, and the subcomplex RPB2-RPB3-RPB10-RPB11-RPB12. The remaining RNAP II subunits, RPB4, RPB6, RPB7, and RPB9, are likely nonessential for RPB1 stability and may be integrated at a later stage. Taken together, these findings show that Hsp90 and R2TP stabilize RPB1 by mediating its interactions with other RNAP II subunits.

R2TP may also be involved in RNAP I and RNAP III assembly since RPAP3-based purifications showed interactions with RPA1 and RPC1, the two largest subunits of RNAP I and RNAP III, respectively [12,16]. Depletion of RPA135, the second largest subunit in RNAP I, increased the interaction between RPA1 and RPAP3, demonstrating that RPAP3 preferentially binds to RPA1 when it is unassembled [12].

## 5. PIKK Complex Assembly and Stabilization

Phosphatidylinositol 3-kinase-related kinases (PIKKs) belong to the Ser/Thr kinase family and are required for cell proliferation, metabolism, and differentiation. The PIKK family is comprised of ATM, ATR, and DNA-PKcs, which are involved in DNA damage sensing, signaling, and repair (Figure 4); mTOR, a central regulator of cell metabolism, growth, and survival (Figure 4); SMG1, involved in nonsense-mediated mRNA decay; and TRRAP, a pseudokinase that lacks catalytic activity but functions as a large protein interaction hub. Although they have diverse functions, PIKKs share a common domain architecture where their N-termini carry long arrays of HEAT repeats [92], and their C-termini phosphorylate target proteins using a region related to the domain of PI3 kinase [93]. The PIKKs oligomerize with other proteins to form complexes, yet none of their binding partners or target proteins are common to all family members. During PIKK complex assembly and function, they each depend on Hsp90, the TTT complex, and the R2TP complex [13,94,95].

### 5.1. Role of Hsp90-TTT in PIKK Complex Assembly

The TTT complex, comprising of TELO2, TTI1, and TTI2, was discovered through a large-scale proteomic analysis identifying Tel2 interactors in fission yeast [96,97]. Each component of the TTT complex is mutually dependent on each other for their stability [98]. TTI1 provides a platform for TELO2 and TTI2 to bind to its central and C-terminal regions, respectively [99]. Functional studies in yeast, *C. elegans*, and mammalian cells show that components of the TTT complex are essential for proper PIKK signaling pathways, namely, the DNA damage response [94,98,100,101,102,103,104,105,106], metabolic stress [107,108,109,110,111,112,113], nonsense-mediated mRNA decay [114,115], and transcriptional regulation [116,117]. The TTT complex recognizes and stabilizes PIKKs cotranslationally before mediating their assembly into larger complexes [95,111,118].

The TTT complex functions as an Hsp90 cochaperone. Co-IP experiments coupled with MALDI-TOF using HeLa S3 cell extracts revealed that FLAG-tagged TTT subunits associate with PIKKs, R2TP subunits, and Hsps [118]. TTI2 was shown to associate with Hsp90 [118]. Co-IPs in HEK293 cells showed that TELO2 and TTI1 could also associate with Hsp90, and Hsp90 ATPase activity was needed to stabilize TELO2 and TTI1 [94]. Thus, Hsp90 ATPase activity is essential for PIKK stabilization and proper PIKK signaling [94,119]. Hsp90 inhibition in HeLa cells interfered with TELO2-ATR and TELO2-mTOR interactions, which decreased the association of ATR with ATRIP, and mTOR with Raptor and Rictor [118]. These findings have important implications on overall cell metabolism. ATR-ATRIP interact with RPA on ssDNA to initiate cell cycle arrest [120]. mTOR and Raptor are part of the mTORC1 complex, which is involved in cell growth and proliferation, and mTOR and Rictor are part of the mTORC2 complex, which is involved in cell survival (Figure 4) [121]. Hsp90 inhibition coupled with glutaminase inhibition has been shown to be an effective therapeutic strategy against mTORC1-driven tumors [122]. Furthermore, Hsp90 inhibition also interfered with TELO2-ATM and TELO2-DNA-PKcs interactions [118]. Similar to ATR and mTOR, ATM and DNA-PKcs interactions with TELO2 may mediate ATM-MRN and DNA-PKcs-Ku70/Ku80 interactions. The ATM-MRN and DNA-PKcs-Ku70/Ku80 complexes are both involved in DNA double-strand break repair (Figure 4) [123,124,125,126].

### 5.2. Role of R2TP-TTT in PIKK Complex Assembly

In addition to Hsp90, the RUVBL1/2 complex also functions as a TTT cochaperone. The mechanism of RUVBL1/2-mediated mTORC1 complex assembly and activation has recently been elucidated. The cryo-EM structure of the human R2TP-TTT complex shows that TTT binds simultaneously with DII domains from consecutive RUVBL1 and RUVBL2 subunits of the RUVBL1/2 hexameric ring [127]. RUVBL1/2 and TTT cooperate to recruit mTOR to the complex. In vitro binding experiments showed that the human TTT complex coprecipitates with a mTOR C-terminal fragment, and the RUVBL1/2 complex coprecipitates with an mTOR N-terminal fragment [127]. Although TTT complex binding to RUVBL1/2 inhibited RUVBL1/2 ATPase activity in vitro [127], RUVBL1 ATPase activity is essential for TTT complex formation and mTOR complex activation *in cellulo* [112]. When endogenous RUVBL1 was knocked out in *TSC2*^−/−^ MEFs, which have high mTORC1-driven translation, and rescued with a RUVBL1 ATPase-activity deficient mutant, mTORC1 complex dimerization was inhibited [112]. RUVBL1 ATPase inhibition prevented mTOR-TELO2, TTI1-TELO2, and RUVBL1-TELO2 interactions, indicating disassembly of the RUVBL1/2-mTOR-TTT complex [112]. Interestingly, piperlongumine, a cancer therapeutic for mTORC1-addicted cells, targets RUVBL1/2 to prevent the formation of the RUVBL1/2-TTT complex [128].

Several scaffold and adaptor proteins mediate RUVBL1/2-TTT-PIKK complex assembly. WAC was identified as an adaptor between RUVBL1/2 and TTT during energy-dependent mTORC1 dimerization [108]. Co-IPs in HeLa cell lysates showed that URI1 associates with TELO2, TTI1, TTI2, Hsp90, RUVBL1, RUVBL2, RPAP3, and PIH1D1 [94], suggesting that it may bridge interactions between TTT, Hsp90, and R2TP. Moreover, PIH1D1 within the R2TP complex acts as an adaptor between RUVBL1/2 and TTT. PIH1D1 binds to two constitutively phosphorylated serine residues (S487 and S491) on TELO2 [13]. Co-IPs using TELO2 knockout MEFs rescued with TELO2 S487A and S491A mutants showed compromised association between TELO2 with PIH1D1 and RUVBL1 [13]. In addition, compared to MEF knockout cells rescued with WT TELO2, cells rescued with the TELO2 substitution mutants had a complete reduction of SMG1, a significant reduction of mTOR, and a minor reduction of ATM, ATR, and DNA-PKcs [13]. Inhibition of CK2, the kinase that phosphorylates TELO2 [13], also reduced levels of endogenous SMG1 [114]. Further analysis of the PIH1D1 binding motif, DpSDD, on TELO2 showed that it was highly conserved from yeast to humans, suggesting a mechanism by which PIH1D1 recognizes its substrates [25]. As mentioned above, the DpSDD motif is present in other proteins involved in PAQosome-mediated assembly pathways [25], including the E3 ligase UBR5, which interacts with the H/ACA ribonucleoprotein complex and regulates ribosomal RNA biogenesis [129]; RNAP II subunit RPB1, the largest subunit in the RNAP II complex; EFTUD2, the splicing component of U5 snRNP; and ECD, an adaptor protein for U5 snRNP complex assembly [130].

## 6. Hsp90- and R2TP-Mediated MRN Complex Stabilization

The MRN complex is involved in sensing, processing, and repairing DNA strand breaks (DSBs). The complex is comprised of the nuclease MRE11, ATPase RAD50, and PIKK scaffold NBS1 (Figure 4). During the DNA damage response, the MRN complex binds to DSBs, recruits and activates the PIKKs ATM and ATR, and facilitates DNA repair by homologous recombination and non-homologous end-joining [131,132,133,134,135]. Hypomorphic mutations in MRE11, NBS1, and RAD50 cause ataxia-telangiectasia-like disease [136], Nijmegen breakage syndrome [137], and Nijmegen breakage syndrome-like disorder [138], respectively. Both ataxia-telangiectasia-like disease and Nijmegen breakage syndrome are characterized by genomic instability, hypersensitivity to radiation, and increased susceptibility to cancer.

MRE11 is a conserved 70–90 kDa dimeric protein that has endo- and exonuclease activity against single- and double-stranded DNA [139,140,141]. MRE11 stability was shown to be dependent on its interaction with PIH1D1 [142]. PIH1D1 interacts with MRE11 at S558/S561 or S688/S689 when both serines of each site are phosphorylated, with the latter being the major binding site [142]. Cells expressing MRE11 mutated at S688/S689 had reduced levels of stable MRE11 compared to WT cells [142]. In addition, RPE1, U2OS, and HCT116 cells treated with siRNA against *PIH1D1* had reduced levels of MRE11 and slightly reduced levels of RAD50 and NBS1 [142].

RAD50 is a 150 kDa protein that contains an ABC-type ATPase domain that binds and unwinds dsDNA termini [143,144]. Hsp90 ATPase activity is essential for RAD50 expression [145]. HO-8910 ovarian cancer cells treated with 17-AAG had significantly reduced levels of RAD50 [145]. Hsp90 is also important for RAD50-mediated BRCA1 recruitment to DSBs. BRCA1, a tumor suppressor protein linked to breast and ovarian cancer, interacts with RAD50 in vitro and in vivo and co-localizes with RAD50, MRE11, and NBS1 in irradiation-induced foci [146]. In MCF7 breast cancer cells, 17-AAG decreased BRCA1 protein levels in a dose- and time-dependent manner and impaired irradiation-induced homologous recombination and non-homologous end joining [147].

NBS1 is an 85 kDa protein containing two BRCT domains that bind pSDpTD motifs on interacting proteins, including repair and checkpoint proteins at DSBs [148,149]. Hsp90α stabilizes NBS1 and ATM, but not MRE11 and RAD50 [150]. Hsp90 also stabilizes the interaction between NBS1 and ATM and is needed for MRN translocation to nuclear foci after irradiation [151]. Upon irradiation-induced ATM activation, ATM phosphorylates both NBS1 and Hsp90α, and pNBS1 dissociates from pHsp90 and translocates to DSBs [150,152]. When PIKKs phosphorylate Hsp90 at Thr 7, Hsp90α also translocates to DSBs [153]. By contrast, another study showed that when ATM phosphorylates Hsp90 at Thr 5 and Thr 7, Hsp90α is not significantly recruited to DSBs [150]. In addition to regulating Hsp90 localization, Hsp90 phosphorylation is essential for MRN stabilization since Cdc7-Dbf4-mediated phosphorylation of S164 on Hsp90 was required for stabilizing the Hsp90-TELO2-MRN complex and resulted in enhanced ATM/ATR signaling [154].

## 7. Hsp90- and R2TP-Mediated TSC Complex Stabilization

The TSC complex, comprising of tumor-suppressor proteins TSC1, TSC2, and TBC1D7, inhibits the mTORC1 complex, which controls cell growth and proliferation (Figure 4) [155]. Loss-of-function mutations in TSC1 or TSC2 have been linked to tuberous sclerosis, a rare genetic disorder that causes tumor growth in multiple organs and neurological symptoms [156]. Within the TSC complex, TSC1 binds and stabilizes both TSC2 and TBC1D7 [157,158,159]. To inactivate mTORC1, TSC2, which contains a GAP domain, catalyzes the conversion of Rheb-GTP to Rheb-GDP [160,161].

TSC1 was reported to be an Hsp90 cochaperone that inhibits Hsp90 ATPase activity (Figure 4) [162]. TSC1 enables TSC2 binding to Hsp90, which prevents TSC2 ubiquitin-mediated proteasomal degradation [162]. TSC1 binding to Hsp90 was also important for stability and activity of kinase client proteins such as c-Src, CDK4, and Ulk1, as well as non-kinase client proteins such as glucocorticoid receptor and folliculin [162]. In bladder cancer cells, TSC1 facilitated Hsp90 acetylation at K407/K419, which increased its binding affinity for Hsp90 inhibitor ganetespib [163]. In contrast to bladder cancer cells, however, CAL-72 and PEER cells, which have a complete loss of TSC1 and reduced TSC2 expression, were also sensitized to ganetespib with IC_50_ values of 22 and 3 nM, respectively [164]. In addition, hepatocellular cancer cell lines SNU-398, SNU-878, and SNU-886, which have a complete loss of TSC2 and normal TSC1 expression, had IC_50_ values of 9, 14, and 35 nM, respectively [164]. Nevertheless, these findings show that TSC1 and TSC2 influence Hsp90 activity.

The TSC complex may be stabilized through the PAQosome. Co-IP experiments in HeLa cells showed that FLAG-tagged URI1 and RPAP3 interacted with endogenous TSC1 and TSC2 [58]. TAP-MS of each TSC complex subunit demonstrated high confidence interactions with RUVBL1, RUVBL2, RPAP3, PIH1D1, WDR92, and URI1 [58]. A SILAC proteomic analysis using the N-terminal domain of PIH1D1 showed that it associated with all three subunits of the TSC complex [59]. The significance of these interactions is unknown. The PAQosome may act as a loading dock that stabilizes each TSC subunit before combining them into a single complex. Moreover, the PAQosome may scaffold TSC1, to regulate Hsp90 ATPase activity, or it may scaffold TSC2, to facilitate loading onto Hsp90 [162].

## 8. Axonemal Dynein Arm Assembly

Motile cilia are small microtubule-based organelles required for fluid transport and cell motility in many organisms. In humans, motile cilia are essential for the generation of left-right asymmetry during embryonic development, sperm motility, and the movement of fluid in the respiratory tract, brain ventricular system, and oviducts [165]. Motile cilia contain a 9 + 2 axoneme comprised of nine outer doublet microtubules and a pair of central microtubules. Between each outer doublet, there are several multiprotein complexes, which include the inner dynein arms (IDA) and outer dynein arms (ODA). Dynein is a AAA+ ATPase that mediates microtubule sliding and subsequent ciliary movement [166]. Before being incorporated into the axoneme, dynein arms are preassembled in the cytoplasm [167,168].

### 8.1. DNAAFs Form Complexes with Hsp90

DNAAFs were discovered through genetic analyses of families with primary ciliary dyskinesia and mutation studies in animals [168,169,170,171,172,173,174,175,176,177,178,179]. Although most of their functions are still being investigated, it is clear that DNAAFs work together with Hsp90 to mediate IDA and ODA assembly [180]. DNAAF2, DNAAF4, DNAAF6, and DNAAF11 have domains that associate with Hsp90, including PIH1, CS, and TPR domains, while DNAAF1, DNAAF3, DNAAF5, and DNAAF7 lack Hsp90 association domains.

In vertebrates, the PIH1 domain is present in at least four proteins: DNAAF2, DNAAF6, PIH1D1, and PIH1D2 [181,182,183]. Each protein has been shown to be involved in ciliary dynein arm assembly [182]. DNAAF2 and DNAAF6 each contain an N-terminal PIH1 domain followed by a CS domain. In mouse testis extracts, DNAAF2 coprecipitated with Hsp70 but not Hsp90 [175], whereas DNAAF6 coprecipitated with both Hsp70 and Hsp90 [181]. In addition, a yeast two-hybrid analysis showed that DNAAF6 interacts with Hsp90, DNAAF2, and DNAAF4 [176].

DNAAF4 contains a C-terminal TPR domain, and a yeast two-hybrid screen showed that it interacts with Hsp70 and Hsp90 C-termini via the EEVD motif that binds TPR domains [184]. These interactions were confirmed through coprecipitation experiments in mouse trachea tissues [177]. In addition, DNAAF4 coprecipitated with DNAAF2 in HEK293 cells [177]. Based on their domains, DNAAF2 and DNAAF4 may form R2TP-like complexes that mediate Hsp90 involvement in dynein arm assembly (Figure 5) [26]. Moreover, TTC12 has recently emerged as another dynein arm assembly factor, and it contains a stretch of three TPR domains [185], suggesting that it may also be involved in forming R2TP-like complexes.

DNAAF1 and DNAAF7, which lack Hsp90 binding domains, have been linked to Hsp90. Streptavidin-II/FLAG tandem affinity purification coupled with mass spectrometry (SF-TAP/MS) experiments using HEK293 lysates showed that DNAAF1 associates with several Hsps, including Hsp70 and Hsp90 [186]. Although DNAAF7 lacks an Hsp90 binding domain, endogenous DNAAF7 coprecipitations from P30 mouse testes, P7 mouse oviducts, and primary ciliated HEK293 cells revealed the presence of Hsp90 [187]. Hsp90 may have an indirect interaction with DNAAF7 through FKBP8, an immunophilin belonging to the FK506-binding protein family, thereby forming a DNAAF7-FKBP8-Hsp90 complex. FKBP8 contains a TPR domain that interacts with Hsp90 [188], and it was present in endogenous DNAAF7 coprecipitations from P30 mouse testes and differentiating human tracheal epithelial cultures [187]. There have been no reports linking DNAAF3 and DNAAF5 to Hsp90. Aside from it being essential for dynein arm assembly, little is known about DNAAF3 function, but it may have a role similar to DNAAF1 and DNAAF2 [168]. Coprecipitation experiments using human bronchial epithelial tissues showed that DNAAF5 does not interact with Hsp70 or Hsp90 [169].

DNAAF11 (formerly named LRRC6) is another essential protein for dynein arm assembly [189,190]. HEK293T cells co-expressing DNAAF7 and DNAAF11 and treated with protein synthesis inhibitor cycloheximide for 48 h had 44.4% of its DNAAF11 remaining, while DNAAF11 expressed alone had 7.8% remaining [191], indicating that DNAAF7 is needed to stabilize DNAAF11. DNAAF11 may interact with Hsp90 directly through its CS domain, or indirectly through its interactors DNAAF7 and RUVBL2 [173,178,187,192]. To release client proteins from Hsp90, p23 binding and ATP hydrolysis is required [193]. Thus, DNAAF11 binding to Hsp90 may promote the release of dynein arms from DNAAF7-FKBP8-Hsp90 to other chaperone complexes, including R2TP and R2TP-like complexes (Figure 5) [187].

### 8.2. R2TP and R2TP-like Complexes Are Dynein Arm Assembly Factors

The R2TP complex may be involved in late-stage dynein arm assembly. Similar to DNAAFs, the catalytic components of R2TP, RUVBL1 and RUVBL2, were demonstrated to be involved in dynein arm assembly through mutational analyses in animal models. Inducible deletion of RUVBL1 in mouse oviducts resulted in the absence of outer dynein arms and the appearance of undefined protein clusters [194]. Streptavidin-II/FLAG tandem affinity purification (SF-TAP) using HEK293 cell lysates showed that DNAAF1 interacts with RUVBL1 and RUVBL2, and that the RUVBL1 interaction was reduced with mutant DNAAF1 [186]. RUVBL1 knockdown in hTERT-RPE1 cells showed increased co-localization between intraflagellar transport protein IFT1 and DNAAF1, suggesting that RUVBL1 mediates DNAAF1 transport or localization [186]. In zebrafish, RUVBL1 and RUVBL2 are enriched in cytoplasmic puncta in zebrafish ciliated tissues, and cilia motility is lost in zebrafish with either RUVBL1 or RUVBL2 mutants [192,195]. RUVBL2 interacts with DNAAF11, which has a similar domain composition to DNAAF1 [192]. The RUVBL2-DNAAF11 complex was essential for dynein arm assembly in zebrafish [192]. Altogether, these findings suggest the presence of cytoplasmic R2TP-like complexes that mediate dynein arm assembly.

In a conditional mouse model, loss of RUVBL1 resulted in immotile spermatozoa due to reduced ODA components, DNAI1 and DNAI2 [195]. In mouse testes, RUVBL2 interacted with Hsp90, suggesting that RUVBL2 scaffolds DNAI1 and DNAI2 to Hsp90 [195]. RUVBL1 may also scaffold IDA and ODA components to Hsp90 through the R2TP-like complex R2SP, comprising of RUVBL1, RUVBL2, SPAG1, and PIH1D2 [196]. Both SPAG1 and RPAP3 contain RPAP3_C and TPR domains, while PIH1D1 and PIH1D2 both contain N-terminal PIH1 and C-terminal CS domains. In zebrafish, SPAG1 null mutations resulted in dorsal body curvature and hydrocephalus, indications of primary ciliary dyskinesia [197], while double-null mutations in PIH1D2 and DNAAF2 resulted in abnormal sperm motility [182]. RUVBL2, SPAG1, and PIH1D2 were found to be ubiquitously expressed in all human tissues and had moderate to high enrichment in the testes [196]. The R2SP complex was shown to facilitate the formation of liprin-α2 complexes [196], which are involved in synaptic vesicle release [198]. Interestingly, PIH1 domain-containing proteins DNAAF2 and DNAAF6 were also enriched in the testes, suggesting the presence of multiple R2TP-like complexes [196].

In addition to R2TP and R2TP-like complexes, proper dynein arm assembly requires WDR92 (recently renamed DNAAF10), which is also highly expressed in human testes [199]. In *Chlamydomonas*, experiments using insertion and truncation mutants showed that WDR92 is needed to stabilize ODA and IDA heavy chains during preassembly [200,201]. Co-IPs using HEK293 cells and in vitro pulldowns showed that WDR92 interacts directly with RPAP3 [200,202], suggesting that a WDR92-R2TP complex is needed for proper dynein arm assembly (Figure 5). In addition, *Drosophila* WDR92 was shown to interact with CG18472, the closest *Drosophila* orthologue of human SPAG1 [197,203]. A proteomic analysis also supports a possible interaction between human WDR92 and SPAG1 [58]. These findings suggest the possibility of a WDR92-R2SP complex.

RUVBL1 and RUVBL2 were recently demonstrated to be involved in the synthesis of cytoplasmic cilia, in which the axoneme is exposed to the cytoplasm [204,205]. Cytoplasmic cilia are found in male gametes, including human and *Drosophila* sperm. While investigating *Drosophila* spermiogenesis, Fingerhut and colleagues identified a novel RNP granule located at the axoneme distal end, the site of ciliogenesis [205]. The RNP granule contained RUVBL1 and RUVBL2, as well as mRNA that encodes axonemal dynein arms. By localizing translation, dynein arms can be integrated into the axoneme directly from the cytoplasm. RUVBL1 and RUVBL2 were essential for dynein arm integration and subsequent spermatozoa motility. Similar to their involvement with other RNPs, RUVBL1 and RUVBL2 were also essential for RNP granule formation [205].

## 9. Concluding Remarks

These studies demonstrate that Hsp90 and R2TP have many diverse and essential roles in macromolecular complex assembly that often complement each other. Before complex assembly, Hsp90 initiates subunit expression (e.g., RAD50, TERT transcription) and regulates RNA levels (e.g., U3 snoRNA, U4 snRNA). During complex assembly, Hsp90 stabilizes clients that compose multisubunit complexes (e.g., L7Ae proteins 15.5K, NHP2, SBP2; RNAP subunits RPA1, RPB1, RPC1; PIKK proteins ATM, ATR, DNA-PKcs, mTOR, SMG-1, TRRAP), while R2TP mediates important interactions between Hsp90 and clients (e.g., Hsp90-RPB1), adaptors and clients (e.g., TELO2-mTOR, TELO2-ATR), and complex subunits (e.g., 15.5K-NOP56, 15.5K-NOP58). After complex assembly, Hsp90 is critical for the function of some complexes (e.g., telomerase elongation, MRN-mediated recruitment of BRCA1 to DSBs), and R2TP together with its associated prefoldin-like module are involved in the localization of assembled protein complexes (i.e., R2TP-mediated Cajal body and nucleolar localization of snoRNPs, URI1-mediated nuclear localization of RNAP II). Thus, in addition to the canonical role of Hsp90 in client stabilization, these findings highlight the additional roles of Hsp90, together with R2TP, in quaternary complex assembly. Determining how Hsp90 integrates its clients into multiprotein complexes may facilitate the discovery of novel therapeutic drug targets. For example, inhibiting Hsp90-mediated TELO2-mTOR interactions may be an effective adjuvant against mTORC1-driven tumors. Thus, the role of Hsp90 during complex assembly and how it functions with its chaperones and cochaperones, especially TTT and R2TP, should be further investigated.

## Figures and Tables

**Figure 1 biomolecules-12-01045-f001:**
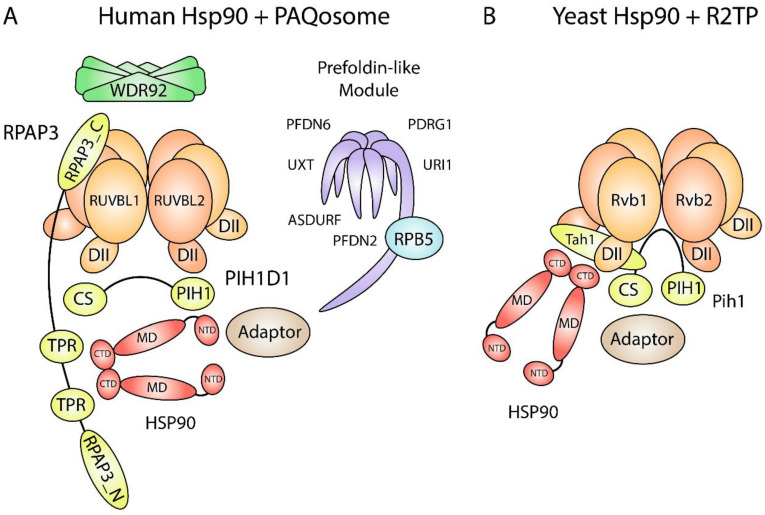
Schematic of Hsp90 and PAQosome subunits. (**A**) Human Hsp90 interacts with R2TP through the TPR domains on RPAP3. The RPAP3 C-terminal domain binds to the ATPase side of RUVBL2 and tethers Hsp90 and PIH1D1 to the rest of the R2TP complex. Some Hsp90 and RUVBL1/2 clients are recruited through adaptors. The PIH1 domain in PIH1D1 binds to proteins that contain a DpSDD/E motif. WDR92 and the prefoldin-like module (UPC) may also act as Hsp90 adaptors since they associate with human R2TP. CS, CHORD domain-containing protein Sgt1 domain; CTD, C-terminal domain; DII, Domain II; MD, middle domain; NTD, N-terminal domain; PIH1, Pih1 homology domain; RPAP3_C, RPAP3 C-terminal domain; RPAP3_N, RPAP3 N-terminal domain; TPR, tetratricopeptide domain. (**B**) Yeast Hsp90 interacts with R2TP through the RPAP3 yeast orthologue Tah1. Tah1 is much smaller than RPAP3, which gives yeast R2TP an open basket structure for client binding. An orthologous prefoldin-like module (UPC) and an orthologue for WDR92 are absent in yeast.

**Figure 2 biomolecules-12-01045-f002:**
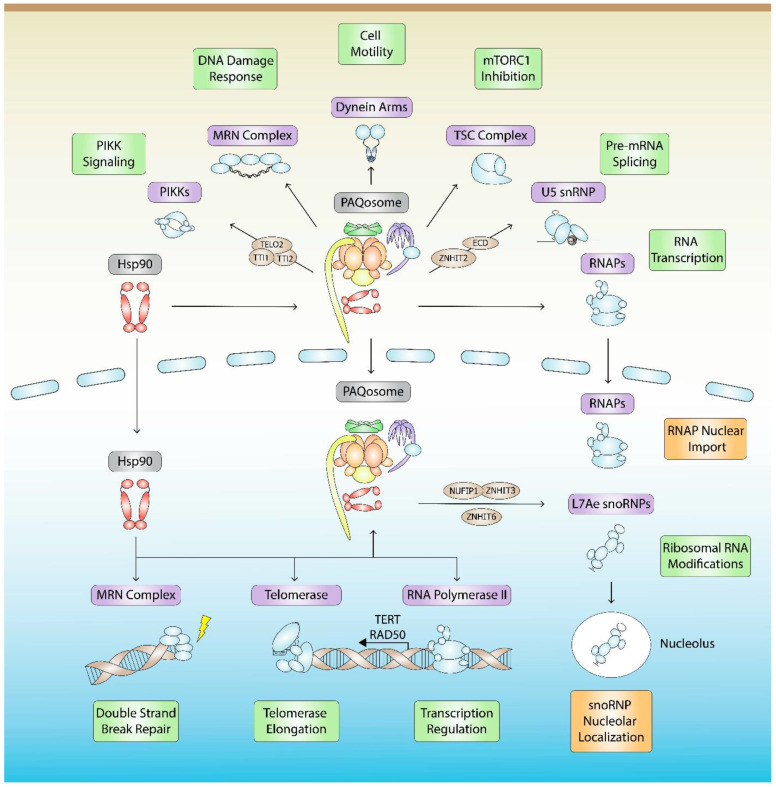
Hsp90- and PAQosome-mediated quaternary assembly and stabilization pathways. Hsp90 together with the PAQosome are involved in the assembly, stabilization, function (green), or localization (orange) of at least seven classes of protein complexes (purple), which include L7Ae snoRNPs, spliceosome snRNPs, RNA Polymerases, PIKKs, MRN, TSC, and dynein arms. RNAs within each RNP complex that are mentioned in the text are listed. R2TP/PAQosome assembly factors (brown) are shown.

**Figure 3 biomolecules-12-01045-f003:**
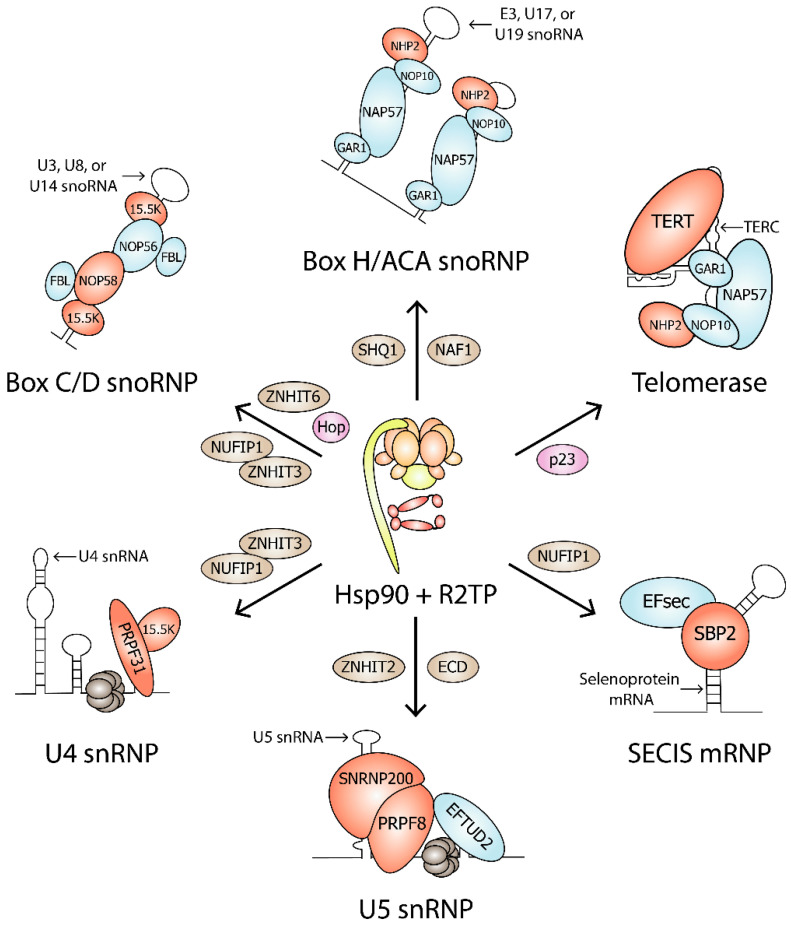
Hsp90 clients in RNP complexes. Hsp90, Hsp90 cochaperones (pink), R2TP, and assembly factors (brown) are involved in the biogenesis of Box C/D snoRNP, Box H/ACA snoRNP, Telomerase, U4 snRNP, U5 snRNP, and SECIS mRNP. Hsp90 clients are shown in red. Protein complex components that are not Hsp90 clients are shown in blue.

**Figure 4 biomolecules-12-01045-f004:**
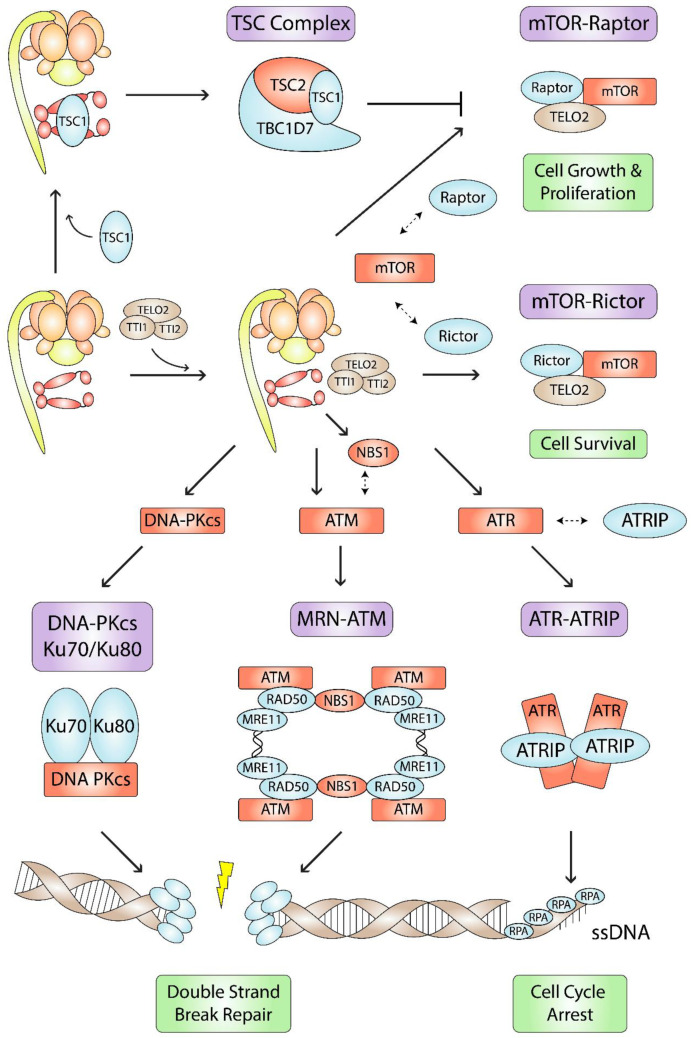
Hsp90- and R2TP-mediated PIKK, MRN, and TSC complex assembly pathways. Hsp90, R2TP, and the TTT (brown) are involved in the assembly, stabilization, or function (green) of several complexes (purple) involved in cell metabolism and DNA damage responses. Hsp90-R2TP stabilizes its clients (red) and mediates interactions (dashed double-sided arrows) between its clients and other complex subunits (blue).

**Figure 5 biomolecules-12-01045-f005:**
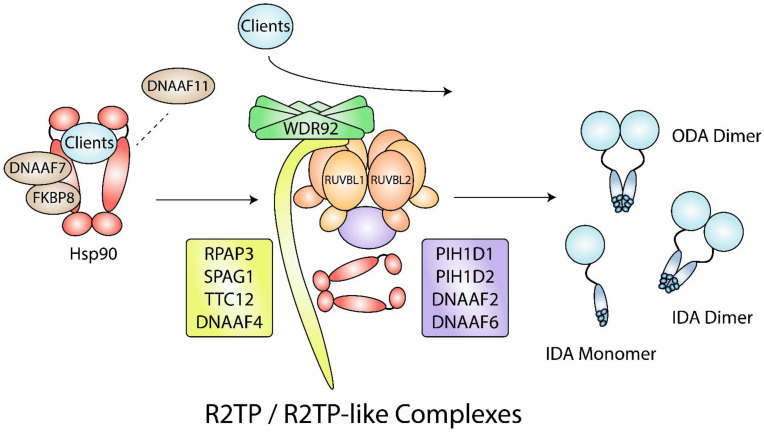
Hsp90- and R2TP/R2TP-like complex-mediated dynein arm assembly. During dynein arm assembly, DNAAF7 and FKBP8 act as Hsp90 cochaperones that are required for the folding of dynein arm clients. DNAAF11 may be needed for client release from Hsp90 to WDR92, R2TP, and R2TP-like complexes. Other clients may not require Hsp90 for folding and may interact with WDR92, R2TP, and R2TP-like complexes directly. R2TP-like complexes contain the RUVBL1/2 hexamer and may have a combination of RPAP3-like (yellow) and PIH1D1-like (purple) proteins. IDA, inner dynein arm; ODA; outer dynein arm.

**Table 1 biomolecules-12-01045-t001:** Nomenclature.

17-AAG	17-(Allylamino)-17-demethoxygeldanamycin
AAA+	ATPases associated with diverse cellular activities
ASDURF	ASNSD1 upstream open reading frame protein
ATM	Ataxia-telangiectasia mutated
ATR	ATM- and RAD3-related
ATRIP	ATR-interacting protein
BRCA1	Breast cancer type 1 susceptibility protein
Cdc7	Cell division cycle 7-related protein kinase
CDK4	Cyclin-dependent kinase 4
CK2	Casein Kinase 2
COPS8	COP9 signalosome complex subunit 8
CS	CHORD domain-containing protein and Sgt1 domain
Cse4	Chromosome segregation protein 4
c-Src	Cellular proto-oncogene tyrosine-protein kinase Src
Dbf4	Protein DBF4 homolog A
DNAAF1	Dynein axonemal assembly factor 1
DNAAF2	Dynein axonemal assembly factor 2
DNAAF3	Dynein axonemal assembly factor 3
DNAAF4	Dynein axonemal assembly factor 4
DNAAF5	Dynein axonemal assembly factor 5
DNAAF6	Dynein axonemal assembly factor 6
DNAAF7	Dynein axonemal assembly factor 7
DNAAF8	Dynein axonemal assembly factor 8
DNAAF11	Dynein axonemal assembly factor 11
DNAI1	Dynein axonemal intermediate chain 1
DNAI2	Dynein axonemal intermediate chain 2
DNA-PKcs	DNA–protein kinase catalytic subunit
ECD	Ecdysoneless homolog
EFTUD2	Elongation factor Tu GTP binding domain containing 2
FKBP8	FK506-binding protein 8
GAR1	Glycine arginine rich protein 1
GPN2	GPN-Loop GTPase 2
GPN3	GPN-Loop GTPase 3
GrinL1A	Glutamate receptor-like protein 1A
Hop	Hsp organizing protein
Hsc82	Heat shock cognate protein 82
Hsp70	Heat shock protein 70
Hsp82	Heat shock protein 82
Hsp90	Heat shock protein 90
IFT1	Interferon-induced protein with tetratricopeptide repeats 1
Ku70	Lupus Ku autoantigen protein p70
Ku80	Lupus Ku autoantigen protein p80
LRRC6	Leucine rich repeat containing 6
MRE11	Meiotic recombination 11
MRN	MRE11-RAD50-NBS1
mRNP	Messenger ribonucleoprotein
mTOR	Mammalian target of rapamycin
mTORC1	Mammalian target of rapamycin complex 1
mTORC2	Mammalian target of rapamycin complex 2
NAF1	Nuclear assembly factor 1
NAP57	Nopp140-associated protein of 57 kDa
NBS1	Nibrin
NHP2	Non-Histone protein 2
NOP10	Nucleolar protein 10
NOP56	Nucleolar protein 56
NOP58	Nucleolar protein 58
NOPCHAP1	NOP protein chaperone 1
NUFIP1	Nuclear FMRP interacting protein 1
PAQosome	Particle for arrangement of quaternary structure
PDRG1	p53 and DNA damage regulated 1
PFDN2	Prefoldin subunit 2
PFDN6	Prefoldin subunit 6
Pih1	Protein interacting with Hsp90
PIH1D1	PIH1 domain-containing protein 1
PIH1D2	PIH1 domain-containing protein 2
PIKK	Phosphatidylinositol-3-kinase-related kinase
Prp19	Pre-mRNA-processing factor 19
PRPF31	Pre-mRNA-processing factor 31
PRPF8	Pre-mRNA-processing-splicing factor 8
R2SP	RUVBL1-RUVBL2-SPAG1-PIH1D2
R2TP	Rvb1–Rvb2–Tah1–Pih1
RAD50	Radiation sensitive 50
Rheb	Ras homolog enriched in brain
RNAP	RNA polymerase
RPA	Replication protein A 70 kDa DNA-binding subunit
RPA1	RNA polymerase I subunit A
RPA135	DNA-directed RNA polymerase I 135 kDa polypeptide
RPAP3	RNA polymerase II-associated protein 3
RPB1	RNA polymerase II subunit B1
RPB2	RNA polymerase II subunit B2
RPB3	RNA polymerase II subunit B3
RPB4	RNA polymerase II subunit B4
RPB5	RNA polymerase II subunit B5
RPB6	RNA polymerase II subunit B6
RPB7	RNA polymerase II subunit B7
RPB8	RNA polymerase II subunit B8
RPB9	RNA polymerase II subunit B9
RPB10	RNA polymerase II subunit B10
RPB11	RNA polymerase II subunit B11
RPB12	RNA polymerase II subunit B12
RPC1	RNA polymerase III subunit C160
RUVBL1	RuvB-like AAA ATPase 1
RUVBL2	RuvB-like AAA ATPase 2
Rvb1	RuvB-like protein 1
Rvb2	RuvB-like protein 2
SBP2	SECIS binding protein 2
SECIS	Selenocysteine insertion sequence
SHQ1	Small nucleolar RNAs of the box H/ACA family quantitative accumulation 1
Sgt1	Suppressor of G2 allele of SKP1 homolog
SMG1	Nonsense-mediated mRNA decay associated phosphatidylinositol-3-kinase-related kinase
snoRNA	Small nucleolar RNA
snoRNP	Small nucleolar ribonucleoprotein
snRNP	Small nuclear ribonucleoprotein
SNRNP200	Small nuclear ribonucleoprotein U5 subunit 200
SPAG1	Sperm-associated antigen 1
Tah1	TPR-containing protein associated with Hsp90
TBC1D7	Tre2-Bub2-Cdc16 domain family member 7
Tel2	Telomere maintenance 2
TELO2	Telomere length regulation protein TEL2 homolog
TERC	Telomerase RNA component
TERT	Telomerase reverse transcriptase
TPR	Tetratricopeptide repeat
Tra1	Transcription-associated protein 1
TRRAP	Transformation/transcription domain-associated protein
TSC	Tuberous sclerosis complex
TSC1	Tuberous sclerosis 1 protein
TSC2	Tuberous sclerosis 2 protein
TTC12	Tetratricopeptide repeat protein 12
TTI1	TEL2 interacting protein 1
TTI2	TEL2 interacting protein 2
TTT	TELO2-TTI1-TT2
UBR5	Ubiquitin protein ligase E3 component N-recognin 5
UPC	Unconventional prefoldin complex
URI1	Unconventional prefoldin RPB5 interactor 1
UXT	Ubiquitously expressed transcript
WAC	WW domain-containing adaptor protein with coiled-coil
WDR92	WD-40 repeat domain 92
ZNHIT2	Zinc finger HIT-type containing 2
ZNHIT3	Zinc finger HIT-type containing 3
ZNHIT6	Zinc finger HIT-type containing 6

## Data Availability

Not applicable.

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
