# Peer review of "The Role of Hsp90-R2TP in Macromolecular Complex Assembly and Stabilization"

_biomolecules, 2022, doi:10.3390/biom12081045_

Round 1

Reviewer 1 Report

Lynham and Houry summarize the interactions of Hsp90-R2TP with various complex components and adoptors. A number of findings from the field are cited, indicating that Hsp90-R2TP has important functions for cells. However, the "role of Hsp90-R2TP in the formation of those complexes" as stated in the title of the paper is not always clearly defined; if a role for Hsp90-R2TP is known other than stabilization of its components, the "additional role" should be more clearly stated in each section . The manuscript is well organized, but the physiological and biological significance of the additional role does not seem to be clearly characterized. At least the following points should be revised prior to publication.

line 116: This section describes the biogenesis of snoRNPs, but does not show any relationship between these detailed biogenesis processes and Hsp90. An introduction to which processes Hsp90 is involved in should be provided in this section.

line 148: Please indicate where Hop is a factor in Figure 3. Figure 3 shows that Hsp90 is "involved" in the assembly of various complexes, but the text discusses where Hsp90 "could be involved" in the detailed complex formation steps. The discussion is too complex for me to understand, so the authors should either illustrate the known complex formation steps and the known interaction between Hsp90 and other factors or clarify the order of assembly and the relationship between the factors. Also, where is the U3 snoRNA shown in Figure 3? Please also illustrate which noncoding RNAs are included in which complexes.

line 182: Ultimately, is the role of Hsp90-R2TP to recruit 15.5K to the NOP56-NOP58 complex correct or not? It seems an unclear statement to me.

line 265: I can't find the references, please unify the citation format

line 346: So, does the HIT domain have no role? Or is there some kind of regulation in place by having multiple combined sites?

line 349: Summarize what the authors think the change in ATPase activity means.

minor points

line 109: Please present rRNA and snoRNA as an unabbreviated form for the first time to facilitate understanding for broad readers.

Author Response

REBUTTAL

Reviewer 1

  1. Lynham and Houry summarize the interactions of Hsp90-R2TP with various complex components and adoptors. A number of findings from the field are cited, indicating that Hsp90-R2TP has important functions for cells. However, the "role of Hsp90-R2TP in the formation of those complexes" as stated in the title of the paper is not always clearly defined; if a role for Hsp90-R2TP is known other than stabilization of its components, the "additional role" should be more clearly stated in each section . The manuscript is well organized, but the physiological and biological significance of the additional role does not seem to be clearly characterized. At least the following points should be revised prior to publication.

line 116: This section describes the biogenesis of snoRNPs, but does not show any relationship between these detailed biogenesis processes and Hsp90. An introduction to which processes Hsp90 is involved in should be provided in this section.

At the end of this intro section, an extra sentence has been added: “During snoRNP biogenesis, Hsp90 stabilizes NOP58, 15.5K, and NHP2 [9].”

  1. line 148: Please indicate where Hop is a factor in Figure 3.

Done

  1. Figure 3 shows that Hsp90 is "involved" in the assembly of various complexes, but the text discusses where Hsp90 "could be involved" in the detailed complex formation steps.

The text has been edited to omit statements such as, “could be involved,” “may be involved,” “could be important for,” etc., unless there is real uncertainty from lack of experiments to date.

  1. The discussion is too complex for me to understand, so the authors should either illustrate the known complex formation steps and the known interaction between Hsp90 and other factors or clarify the order of assembly and the relationship between the factors.

To help clarify this discussion, the paragraph from lines 138-152 has been broken into two paragraphs, each with additional information. The first paragraph highlights how Hsp90 affects snoRNA synthesis, however, a detailed mechanism is not yet available, but the text alludes to Hsp90 somehow affecting box C/D snoRNP biogenesis. The second paragraph clarifies this by suggesting that Hsp90 stabilizes NOP58 through the Hsp70-Hop-Hsp90 pathway.

  1. Also, where is the U3 snoRNA shown in Figure 3? Please also illustrate which noncoding RNAs are included in which complexes.

snoRNAs and all other RNAs have been labelled in Figure 3.

  1. line 182: Ultimately, is the role of Hsp90-R2TP to recruit 15.5K to the NOP56-NOP58 complex correct or not? It seems an unclear statement to me.

Hsp90 is needed for both NOP58 and 15.5K stability. Hsp90 and its associated chaperones stabilize NOP58 while R2TP recruits 15.5K to NOP56-NOP58. This has been made clearer in the revised manuscript. Inserted parts are underlined:

“Hsp90 inhibition in HEK293 cells resulted in the disappearance of both NOP58 and 15.5K [9], but the link between Hsp90 ATPase activity and 15.5K stabilization remains unclear. Hsp90 may indirectly stabilize 15.5K by stabilizing NOP58 first. Rather than interacting with Hsp90, 15.5K can bind RUVBL1, RUVBL2, and RUVBL1/2 in the presence of ATP [55,56]. Additionally, RUVBL1/2 was shown to bridge the interaction between 15.5K and core proteins NOP56 and NOP58 [55], which may be important for 15.5K stability. Taken together, Hsp70, Hop, Hsp90, R2TP, and NOPCHAP1 stabilize NOP58, and RUVBL1/2 subsequently recruits 15.5K to NOP56-NOP58, thereby stabilizing 15.5K.

  1. line 265: I can't find the references, please unify the citation format

Fixed

  1. line 346: So, does the HIT domain have no role? Or is there some kind of regulation in place by having multiple combined sites?

Added a sentence that clarifies a potential role for the HIT domain: “Rather than mediating the RUVBL1/2-ZNHIT2 interaction, the HIT domain may regulate the conformation and nucleotide state of RUVBL1/2 [85].”

  1. line 349: Summarize what the authors think the change in ATPase activity means.

“Interestingly, the intrinsically low ATPase activity of RUVBL1/2 hexamers with one walker B mutant, in either RUVBL1 or RUVBL2, was significantly increased with ZNHIT2 present, suggesting that ZNHIT2 affects the activity of both RUVBL1 and RUVBL2 subunits [85].”

  1. minor points

line 109: Please present rRNA and snoRNA as an unabbreviated form for the first time to facilitate understanding for broad readers

Done

Reviewer 2 Report

This review by Walid and Jeffrey, reports the structure and function of the R2TP-HSP90 complex 

and provide a nice evolution overview of the co-chaperone complex. As an expert in the structural biology of human R2TP, this review gives important insights into the arrangement and activation of diverse cellular complexes by R2TP-HSP90.

I would like to see this published, as it has certain impact, is well

performed, and is well written. However, I was shocked by the openness manner

in which some of the signalling PIKKs complexes are presented in Figure 4.  For instance, mTOR complexes containing PRAS40 or DEPTOR. Figure 4 could mislead a role of R2TP in signalling which is has not been described so far. I would like minor revisions in order to make the manuscript more conservative in that part, far from any signalling pathway. However, this should not prevent publication.

Author Response

REBUTTAL

Reviewer 2

  1. This review by Walid and Jeffrey, reports the structure and function of the R2TP-HSP90 complex and provide a nice evolution overview of the co-chaperone complex. As an expert in the structural biology of human R2TP, this review gives important insights into the arrangement and activation of diverse cellular complexes by R2TP-HSP90. I would like to see this published, as it has certain impact, is well performed, and is well written. However, I was shocked by the openness manner in which some of the signalling PIKKs complexes are presented in Figure 4. For instance, mTOR complexes containing PRAS40 or DEPTOR. Figure 4 could mislead a role of R2TP in signalling which is has not been described so far. I would like minor revisions in order to make the manuscript more conservative in that part, far from any signalling pathway. However, this should not prevent publication.

We agree with this comment, as subunits of the mTORC1 and mTORC2 (other than RAPTOR and RICTOR) complexes have not been implicated in Hsp90/R2TP assembly and stabilization pathways. In Figure 4, mTORC1 and mTORC2 have been replaced by mTOR-RAPTOR-TELO2 and mTOR-RICTOR-TELO2, respectively. In addition, the paragraph starting at line 431 has been expanded to better explain the pathways in Figure 4.

Reviewer 3 Report

The provided review focusses on Hsp90 and the R2TP and its implications in macromolecular  complex formation.  This manuscript provides an in depth history and analysis of how Hsp90 functional complexes participates in several macromolecular assembly.  However, a few issues that were picked that needs to be fixed to improve the quality of this manuscript are listed below in point format.

Page 1 line 11,  are all Hsp90 essential in all organisms if not the first sentence should be revised to indicate such?

Page 9, figure 2, add a comment in main text on the translocation of PAQosome, does it traverse the nuclear pores as a complex or it re assembles in the nucleus.

Page 11 line 147, clarify what is the role of Hop on NOP58 wildtype form is as that will be the physiological form with reference to the NOP58-A283P mutant.

Page 12 line 222, add a comment or disclaimer as Geldanamycin may have other binding targets that affect the viability of proteins such as Hsp90 clients. In addition, there seems to be blurred cross reference to the different Hsp90 isoforms. Which Hsp90 was the target in these studies cited and does this mean all Hsp90s are the same?

Page 13 line 269-284, use the correct referencing format for consistency.

Page 14 line 298,  why Hsp70 is coming in without introduction, also comment on the implication of these if these findings are due to limited folding competence?

Page 15 line 385, add a comment on the effect of geldanamycin on the structure of Hsp90 if known, or at first when introducing the effect of the inhibitor.

Page 19 line 502, briefly mention the cell types in addition to the acronym.

Page 22 line 607, use the acronym for heat shock proteins (Hsp).

Author Response

REBUTTAL

Reviewer 3

  1. The provided review focusses on Hsp90 and the R2TP and its implications in macromolecular complex formation. This manuscript provides an in depth history and analysis of how Hsp90 functional complexes participates in several macromolecular assembly.  However, a few issues that were picked that needs to be fixed to improve the quality of this manuscript are listed below in point format.

Page 1 line 11,  are all Hsp90 essential in all organisms if not the first sentence should be revised to indicate such?

Hsp90 is essential in all eukaryotes and some bacteria. Including this detail, however, to clarify what is meant by “essential” would not add anything significant to main message of the abstract, thus, we have opted to replace “essential” with “ubiquitous.”

  1. Page 9, figure 2, add a comment in main text on the translocation of PAQosome, does it traverse the nuclear pores as a complex or it re assembles in the nucleus.

The details of PAQosome localization have not been determined. But one group has suggested that URI1 mediates nuclear/cytoplasmic shuttling of RNAP II subunits as part of the PAQosome. This has been added in the text: “Moreover, URI1 mediates nuclear and cytoplasmic shuttling of RNAP subunits, and it has been suggested to do so as part of the PAQosome [20,21]. Thus, PAQosome assembly may occur in the cytoplasm with URI1 facilitating its transport into the nucleus and vice-versa (Figure 2).”

  1. Page 11 line 147, clarify what is the role of Hop on NOP58 wildtype form is as that will be the physiological form with reference to the NOP58-A283P mutant.

The likely role of Hop on NOP58 has been included, “Therefore, NOP58 is likely stabilized through the Hsp70-Hop-Hsp90 pathway during its maturation and assembly into snoRNPs.”

  1. Page 12 line 222, add a comment or disclaimer as Geldanamycin may have other binding targets that affect the viability of proteins such as Hsp90 clients.

A disclaimer has been added, “A caveat to consider is that geldanamycin may have additional binding targets that affect the viability of Hsp90 clients.”

  1. In addition, there seems to be blurred cross reference to the different Hsp90 isoforms. Which Hsp90 was the target in these studies cited and does this mean all Hsp90s are the same?

Hsp90α or Hsp90b have almost identical structure and function. Most studies do not differentiate between the two. A brief note has been included in the first paragraph: “Hsp90 isoforms (referred to as Hsp90 herein) exist as dynamic homodimers, with each protomer …” In addition, a disclaimer has been included at the end of section 1: “Of note, human “Hsp90” in these studies may refer to either isoform, Hsp90α or Hsp90b since they have nearly identical structural and functional similarities that cannot be easily distinguished from one another.”

  1. Page 13 line 269-284, use the correct referencing format for consistency.

Done

  1. Page 14 line 298, why Hsp70 is coming in without introduction, also comment on the implication of these if these findings are due to limited folding competence?

Hop and Hsp70 have been added into the introduction section: “Hsp90 client loading is largely dependent on Hsp70, which binds to nascent or partially folded polypeptides with exposed hydrophobic residues [5,6], and Hop, which functions as an adaptor between Hsp70 and Hsp90 [7].”

  1. Page 15 line 385, add a comment on the effect of geldanamycin on the structure of Hsp90 if known, or at first when introducing the effect of the inhibitor.

The mechanism of geldanamycin is briefly included: “HEK293 cells expressing rat U3 snoRNA and treated with geldanamycin, an Hsp90 inhibitor that blocks the ATP binding site [2,48], had less U3 snoRNA accumulation [9].”

Going into further detail about geldanamycin would detract from what is being discussed in this section.

  1. Page 19 line 502, briefly mention the cell types in addition to the acronym

“HO-8910 ovarian cancer cells treated with 17-AAG had significantly reduced levels of RAD50 [145].”

  1. Page 22 line 607, use the acronym for heat shock proteins (Hsp).

Done

Round 2

Reviewer 1 Report

The revision made it easier to understand.

Author Response

Thanks.